# Post-Treatment with Erinacine A, a Derived Diterpenoid of *H. erinaceus*, Attenuates Neurotoxicity in MPTP Model of Parkinson’s Disease

**DOI:** 10.3390/antiox9020137

**Published:** 2020-02-04

**Authors:** Kam-Fai Lee, Shui-Yi Tung, Chih-Chuan Teng, Chien-Heng Shen, Meng Chiao Hsieh, Cheng-Yi Huang, Ko-Chao Lee, Li-Ya Lee, Wan-Ping Chen, Chin-Chu Chen, Wen-Shih Huang, Hsing-Chun Kuo

**Affiliations:** 1Department of Pathology, Chang Gung Memorial Hospital, Chiayi 61363, Taiwan; lkf2002@cgmh.org.tw; 2Department of Hepato-Gastroenterology, Chang Gung Memorial Hospital, Chiayi 61363, Taiwan; ma1898@adm.cgmh.org.tw (S.-Y.T.); ccteng@gw.cgust.edu.tw (C.-C.T.); gi2216@adm.cgmh.org.tw (C.-H.S.); 3College of Medicine, Chang Gung University, Taoyuan 33302, Taiwan; wen1204@cgmh.org.tw; 4Department of Nursing, Chang Gung University of Science and Technology, Chiayi 61363, Taiwan; 5Division of Colon and Rectal Surgery, Department of Surgery, Chang Gung Memorial Hospital, Chiayi 61363, Taiwan; chris0912@cgmh.org.tw (M.C.H.); bluesky@cgmh.org.tw (C.-Y.H.); 6Graduate Institute of Clinical Medical Sciences, College of Medicine, Chang Gung University, Taoyuan 33302, Taiwan; 7Division of Colorectal Surgery, Department of Surgery, Chang Gung Memorial Hospital, Kaohsiung Medical Center, Chang Gung University College of Medicine, Kaohsiung 83301 Taiwan; kclee@cgmh.org.tw; 8Department of Information Management & College of Liberal Education, Shu-Te University, Kaohsiung 82445, Taiwan; 9Grape King Biotechnology Inc (Grape King Bio Ltd.), Zhong-Li, Taoyuan 32542, Taiwan; ly.lee@grapeking.com.tw (L.-Y.L.); wp.chen@grapeking.com.tw (W.-P.C.); gkbioeng@grapeking.com.tw (C.-C.C.); 10Research Center for Industry of Human Ecology, Chang Gung University of Science and Technology, Taoyuan 33303, Taiwan; 11Chronic Diseases and Health Promotion Research Center, CGUST, Chiayi 61363, Taiwan

**Keywords:** *Hericium erinaceus*, MPTP, ROS, PAK1, p21

## Abstract

*Hericium erinaceus*, a valuable pharmaceutical and edible mushroom, contains potent bioactive compounds such as *H. erinaceus* mycelium (HEM) and its derived ethanol extraction of erinacine A, which have been found to regulate physiological functions in our previous study. However, HEM or erinacine A with post-treatment regimens also shows effects on 1-methyl-4-phenyl-1,2,3,6-tetrahydropyridine (MPTP)-induced neurotoxicity, but its mechanisms remain unknown. By using annexin-V–fluorescein-isothiocyanate (FITC)/propidium iodide staining and a 2’,7’ –dichlorofluorescin diacetate (DCFDA) staining assay, the cell death, cell viability, and reactive oxygen species (ROS) of 1-methyl-4-phenylpyridinium (MMP^+^)-treated Neuro-2a (N2a) cells with or without erinacine A addition were measured, respectively. Furthermore, signaling molecules for regulating the p21/GADD45 cell death pathways and PAKalpha, p21 (RAC1) activated kinase 1 (PAK1) survival pathways were also detected in the cells treated with MPP^+^ and erinacine A by Western blots. In neurotoxic animal models of MPTP induction, the effects of HEM or erinacine A and its mechanism in vivo were determined by measuring the TH-positive cell numbers and the protein level of the substantia nigra through a brain histological examination. Our results demonstrated that post-treatment with erinacine A was capable of preventing the cytotoxicity of neuronal cells and the production of ROS in vitro and in vivo through the neuroprotective mechanism for erinacine A to rescue the neurotoxicity through the disruption of the IRE1α/TRAF2 interaction and the reduction of p21 and GADD45 expression. In addition, erinacine A treatment activated the conserved signaling pathways for neuronal survival via the phosphorylation of PAK1, AKT, LIM domain kinase 2 (LIMK2), extracellular signal-regulated kinases (ERK), and Cofilin. Similar changes in the signal molecules also were found in the substantia nigra of the MPTP, which caused TH+ neuron damage after being treated with erinacine A in the post-treatment regimens in a dose-dependent manner. Taken together, our data indicated a novel mechanism for post-treatment with erinacine A to protect from neurotoxicity through regulating neuronal survival and cell death pathways.

## 1. Introduction

Parkinson’s disease (PD), one of the most common adult-onset movement disorders worldwide [1], is frequently categorized as a leading cause of age-associated movement disorder. Death due to progressive nigrostriatal dopaminergic neurodegeneration, which arises from various environmental and genetic factors, usually results in the consequent loss of projection fibers in the striatum [2]. The pathology of Parkinson’s disease has the features of rapid progression and a poor outcome [3], which are major problems affecting PD therapy; however, current clinical dopamine (DA) replacement therapy for patients with Parkinson’s disease provides temporary symptomatic relief but fails to cease disease progression. Thus, it is imperative to develop more effective drugs to treat the progression of Parkinson’s disease.

One of the well-established PD animal models is based on implanting 1-methyl-4-phenyl-1,2,3,6-tetrahydropyridine (MPTP) into animals in order to produce the neurotoxin 1-methyl-4-phenylpyridinium (MPP^+^), which, similar to PD, causes permanent symptoms of Parkinson’s disease by destroying dopaminergic neurons in the substantia nigra of the brain [4]. Basically, the initial biochemical event of the neurotoxic action of MPTP is two-step oxidation by monoamine oxidase B in glial cells to form MPP^+^ [5]. Following uptake by the synaptic dopamine reuptake system, the MPP^+^ is further concentrated by the electrical gradient of the inner membrane and then slowly penetrates the hydrophobic reaction site on the NADH dehydrogenase. The MPP^+^ combined with NADH dehydrogenase leads to the cessation of oxidative phosphorylation, ATP depletion, and cell death. On the other hand, MPP^+^ treatment generates reactive oxygen species (ROS) in dopaminergic neurons via a two-wave NADPH-oxidase-dependent cascade [6]. Indeed, excessive free radical generation (ROS and reactive nitrogen species (RNS)) is the key event causing progressive neuronal damage in numerous diseases of the nervous system, including Parkinson’s disease [7,8]. Therefore, MPTP-induced dopamine neuron injury has been widely used in rodent models for the development of replacement intervention agents. Targeting the reduction of free radicals from the mitochondria and chronic neuroinflammatory responses are considered valid therapies for PD [9]. The accumulation of ROS induces the formation of the inositol-requiring enzyme 1 α/TNF Receptor Associated Factor 2 (IRE1α/TRAF2) complex and the signaling pathways of JNK1/2, p38, and nuclear factor kappa-light-chain-enhancer of activated B cells (NF-κB), resulting in neuron inflammation and neuron death [10]. Therefore, the elimination of ROS-mediated signaling is considered a candidate target for the treatment of neurodegenerative diseases.

On the other hand, dysregulation of the signaling pathways that involve neural development, growth, and survival occurs during progressive neurodegeneration [11]. For instance, the oxidative-stress-induced downregulation of PAK1 activity could be involved in the loss of mesencephalic DA neurons through the modulation of neuronal death [12]. Its dysregulation is also associated with the pathogenesis of neurodegenerative diseases, such as Alzheimer’s disease (AD) and Huntington’s disease (HD), as well as mental retardation [13]. PAK1 knockout exhibits defects in brain development [14]. Under the MMP^+^ challenge, unusual AKT and MEK signals promote BAD-mediated neuron death [15]. In addition, the LIM domain kinase 2 (LIMK2) signaling pathway induced by the actin cytoskeleton regulatory RhoC promotes cells survival. By contrast, LIMK2-dysregulated DRP1-mediated mitochondrial fission induces programmed necrotic neuronal death [16]. A better understanding of the regulation of neural survival signals may provide new insights into drug screening for the prevention of progressive neurodegeneration.

*Hericium erinaceus* (Lion’s mane or Yamabushitake) is an edible mushroom with an extensively documented range of therapeutic properties [17]. Erinacine A, one of the active diterpenoid compounds isolated from the cultured mycelium of the *Hericium erinaceus*, exerts numerous biological properties [18], including antioxidant activity [19], hypolipidemic activity [20], hemagglutinating activity [21], antimicrobial activity [22], antiaging activity [23], immune modulation, anticancer activities [24], and neuroprotection [25]. Our previous study demonstrated that both pretreatment of *H. erinaceus* mycelium (HEM) and erinacine A through oral intake can protect against MPTP-induced neurotoxicity through the inhibition of endoplasmic reticulum (ER)-stress-mediated cell apoptosis in vivo. [25,26] However, whether erinacine A with post-treatment regimens is a valid therapeutic agent for treating neurodegenerative disease remains unclear. In this study, our data indicated that post-treatment with erinacine A prevents MPTP-induced neurotoxicity through increasing the neuronal survival pathways of PAK1, AKT, LIMK2, MEK, and Cofilin and by reducing the cell death pathways of IRE1α, inositol-requiring enzyme 1 α (IRE1α); TNF Receptor Associated Factor 2 (TRAF2), Apoptosis signal-regulating kinase 1 (ASK1), growth Arrest and DNA Damage (GADD45), and p21.

## 2. Materials and Methods

### 2.1. *Hericium erinaceus* Extracts and Analysis of Erinacine A

Fresh dried mycelium of *H. erinaceus* (2 kg) was extracted using 95% ethanol. The extracted ethanol solution was concentrated and fractionated by solvent partition between ethyl acetate (EtOAc) and water to afford an H_2_O layer and EtOAc layer. The EtOAc layer analysis was subjected to silica gel column chromatography according to previous studies [16,24], while the HPLC analysis of erinacine A was performed with minor modifications. The analytical column was a COSMOSIL 5C18-AR-II (250 × 4.6 mm; particle size 5 μm, Nacalai USA, Inc., Kyoto, Japan). The 5 mg/kg erinacine A in the *H. erinaceus* extracted with 85% ethanol was confirmed and quantified by HPLC. The chemical compounds suggested in this article, erinacine A (PubChem CID: 10410568), the HPLC chromatogram (as supporting material), and the calibration curve used are shown in Figure 1 [27].

### 2.2. Animals

C57BL/6 mice aged 8–10 weeks were kept individually in a cage with free access to water and food and lived in a 12 h light/12 h darkness cycle. Animal care and the general protocols for animal use and MPTP experiments were approved by the Institutional Animal Care and Use Committee of Chang Gung Memorial Hospital (IACUC Approval No: 2017031401). There were four treatment groups of animals, including a sham control group (I), an MPTP group (II), an erinacine A group (III, 1 mg/kg,) and two *H. erinaceus* wet mycelia (HEM) groups (III, 10.76 mg and IV, 21.52 mg). Accordingly, the mice were intraperitoneally (i.p.) injected with MPTP-HCl (30 mg/kg; Sigma, St. Louis, MO) (the MPTP group) or saline (the control group) over 4 days. After the first MPTP injection, the mice received HEM (dissolved in H_2_O; HEM groups) with oral administration or erinacine A (dissolved in dimethyl sulfoxide (DMSO); erinacine A groups) with intraperitoneal administration or an equivalent volume of saline (sham-operated group) for an additional 5 days. The mice were sacrificed 8 days after MPTP injection and their brains were collected for further analysis [25].

### 2.3. Chemical Reagents and Antibodies

The antibodies used in this study were as follows: anti-tyrosine phospho-hydroxylase (Ser31), anti-GAPDH, anti-â-actin, anti-4-hydroxy-2-nonenal (4-HNE), anti-nitro-tyrosine, anti-CHOP, anti-Fas, anti-Bax, anti-NFêB, anti-p65, anti-histone H1, anti-TRAF2, anti-IRE1á, and anti-phospho-IKB-â, all of which were purchased from Santa Cruz Biotechnology (Santa Cruz, CA, USA). The anti-phospho-p38 anti-MAPK (Thr180/Tyr182) and anti-phospho-JNK1/2 (Thr183/Tyr185) were obtained from Cell Signaling Technology (Beverly, MA, USA). The TdT-mediated dUTP Nick End Labeling (TUNEL) kits, MPP^+^, sodium dodecyl sulfate (SDS), NP-40, sodium deoxycholate, and the protease inhibitor cocktail were purchased from Roche (Germany) and Sigma (St. Louis, MO, USA), respectively.

### 2.4. Immunohistochemistry (IHC)

After chloral hydrate administration, the treated mice were sacrificed by decapitation 8 days after MPTP treatment. The removed brains were placed in an ice-cold saline solution for 10 min and then immediately fixed in 10% formalin overnight. The brain sections (seven 4 ìm thick transverse slides) with neuron impairment areas were dehydrated with graded ethanol (10%, 30%, 50%, 70%, and 100%), passed through chloroform, embedded in paraffin, and then assessed by hematoxylin and eosin (H&E) staining. IHC staining was performed in the paraffin sections of the striatum and substantia using monoclonal rabbit antibodies (1:100 dilution) against tyrosine hydroxylase, Ask1, p21, p-PAK1, p-cdc42, p-PI3K p110, p-AKT, and p-LIMK2 along with a biotinylated secondary antibody (Vectastain Universal Elite ABC Kit, Foster City, CA, USA). The primary and secondary antibodies were incubated for 1 h at 25 °C. The omission of the primary antibodies served as a negative control. Digital images were captured and the expression levels of the proteins (brown color) were quantitatively evaluated from three randomly selected microscopic fields (200× magnification) on each slide by using an Olympus Cx31 microscope (Canon A640) with an Image-pro Plus medical image analysis system. The average integral optical density (AIOD = positive area × optical density/total area) was indicated as the IHC index [28].

### 2.5. Cell Culture

The mouse N2a (Neuro-2a) cells and mouse neuron substantia nigra cells were taken from the American Tissue Culture Collection (ATCC, Manassas, VA, USA) and ScienCell Research Laboratories (Carlsbad, CA, USA), respectively. The mouse N2a cells were cultured in a medium consisting of human normal astrocytes (HNA). The neuron substantia nigra cells were cultured in Dulbecco’s Modified Eagle Medium (DMEM, Gibco) supplemented with 10% fetal calf serum (Gibco), nonessential amino acids, 1 mM sodium pyruvate, and 1% antibiotics (100 units/mL of penicillin and 100 ìg/mL of streptomycin). All cells were grown in plastic tissue culture flasks, dishes, or microplates (Nunc, Naperville, Denmark) and were incubated at 37 °C in a humidified atmosphere of 5% CO_2_ and 95% air [29].

### 2.6. Assessment of Cell Viability and Apoptosis Assay

An MTT quantitative colorimetric assay was used to measure cell viability. The cells were seeded at a density of 2 × 10^4^ cells/mL and incubated with MPP^+^ for 24 h. After treatment, the cells were replaced with the MTT medium (0.5 mg/mL) and incubated for 4 h. Following solubilization with isopropanol to produce formazan, the number of viable cells was spectrophotometrically and proportionally measured at 563 nm. Annexin V/propidium iodide (PI) (Biosource International, San Diego, CA, USA) with flow cytometric analysis (FACSCalibur, Becton Dickinson, Missouri, TX, USA) and CellQuest software (BD CellQuest™ software, San Jose, CA, USA) were used to measure cell apoptosis (V+/PI−). The data represented three independent experiments and were analyzed with CellQuest and WinMDI software (WinMDI™ software, San Diego, CA, USA) [30].

### 2.7. Preparation of Total Cell Extracts and Immunoblotting Analysis

A buffer (1% NP-40; 0.5% sodium deoxycholate; 0.1% SDS; and a protease inhibitor mixture of phenylmethylsulfonyl fluoride, aprotinin, and sodium orthovanadate) was used to lyse the cells. After protein quantification, the total cell lysate (50 ìg of protein) was separated by SDS-polyacrylamide gel electrophoresis (PAGE) (12% running, 4% stacking). After member transfer, designated antibodies were used for immunoblotting. The signals of the blots were analyzed by the Western-Light chemiluminescent detection system (Bio-Rad, Hercules, CA, USA), as previously described [31].

### 2.8. Statistical Analyses

All data are expressed as the mean ± standard deviation (SD) of three independent experiments and were analyzed by one-way analysis of variance (ANOVA) using the SAS software statistical package SigmaPlot, version 9.0 (SAS Institute Inc., Cary, NC, USA) [25].

## 3. Results

### 3.1. Erinacine A Prevention of MPP^+^-Induced Cell Death and ROS Generation of N2a Cells

Our previous study demonstrated that *H. erinaceus* mycelium and its structural analog erinacine A have nerve-growth properties that allow them to aid in the prevention of ischemic injury and brain impairment in a mouse model that resembles PD to neurons in the central nervous systems of subjects undergoing excessive oxidative stress [17,25]. Based on previous studies, we assayed whether erinacine A, a derivative of *H. erinaceus*, was capable of protecting neurons. N2a cells from a mouse neural-crest-derived cell line were cultured. MMP^+^, a neurotoxin, was used for the depletion of ATP and eventual cell death. Our results showed that the MMP^+^ treatment induced the condensed and fragmented nuclei of N2a cells (Figure 2A), indicating that MPP^+^ caused apoptotic N2a cells. The addition of erinacine A at 1 and 5 µM allowed the rescue of cell apoptosis by MMP^+^ in the N2a cells (Figure 2A). Similarly, the flow cytometry analysis of the annexin V and PI double staining revealed that erinacine A recovered MMP+-induced necrotic (PI-positive cells) and apoptotic (annexin-V-positive cells) cells. As shown in Figure 2B, MMP^+^ induced significant cell death, determined by annexin V-FITC/PI staining, which was quantified as a percentage of annexin-V-positive cells and shown as 49%. Erinacine A treatment of the N2a cells also resulted in increased neuron survival by 14% and 22%, respectively. The extent of apoptosis was 35% and 27% in the indicated erinacine A treatment of the N2a cell groups, respectively. To further study if erinacine A could inhibit ROS production in order to protect neurons, FACS analysis, which is useful for H2DCFDA as an indicator of ROS, showed that MPP^+^ treatment increased ROS by ~1.7-fold, which was reversed by cotreatment with erinacine A (Figure 2C). Together, the data indicated that erinacine A rescued the MMP^+^-induced N2a cell damage through the inhibition of ROS production by 1.3-fold.

### 3.2. Erinacine A Reduction of MMP+ and Increased Interaction of TRAF2 and IRE1, as well as Expression of GADD45 and p21

Some intercellular regulators play important roles in ROS-mediated progressive cell death, including TRAF2, IRE1, and GADD45 [32]. When ROS accumulation is increased in the cells, the IRE1α/TRAF2 interaction increases and activates the downstream molecular GADD45, ASK1, and p21, leading to neuron death [10]. Our data showed that MMP+ treatment increased the interaction of TRAF2 and IRE1, as well as the expression of GADD45 and p21. (Figure 3). However, the increase in the interaction and the GADD45 and p21 levels were reduced by the treatment of erinacine A. The data suggested that erinacine A inhibited the N2a cells via inhibiting the ROS-mediated IRE1α/TRAF2-GADD45-p21 signaling.

### 3.3. Erinacine A Enhanced Phosphorylation of PAK1/LIMK2/MEK/Cofilin under MPP^+^ Challenge

Our results clearly showed that MPP^+^ resulted in ROS-mediated neuron cell death and that oxidative damage was relative to the expression of the IRE1α/TRAF2-GADD45-p21 triggering pathways, which may function downstream of the ER stress to induce apoptosis in response to neuronal damage. On the other hand, we determined if the erinacine A treatment also affected survival mediators such as PAK1, LIMK2, MEK, and Cofilin in order to protect neurons under MMP^+^ challenge. The Western blot analysis revealed that the erinacine A treatment increased the phosphorylation status of PAK1, LIMK2, MEK, and Cofilin (Figure 4). Conversely, treatment with MPP^+^ resulted in the significant inhibition of AK1, LIMK2, MEK, and Cofilin phosphorylation after 24 h. The data suggested a novel role of erinacine A in the induction of neuronal survival pathways.

### 3.4. Post-Treatment with *H. Erinaceus* Mycelium and Its Structural Analog Erinacine A Weakened MPTP-Induced Neurotoxins and Movement Deficits in Mice

Next, we determined if erinacine A i.p. and oral administrations of HEM could be used as therapeutic agents for PD to protect DA neurons in an MPTP-induced PD model of motor impairment and neurotoxicity (Figure 5). The animals were i.p. injected with MPTP, and 5 days later, *H. erinaceus* mycelium and erinacine A were introduced into the animals at a concentration of 30 mg/kg (Figure 5). In another 5 days, the accelerating rotarod test and IHC staining for TH protein were performed to detect motor activity and the number of DA neurons for TH (Figure 5). Our data showed that the MMP^+^ treatment decreased the maximum time on rod that can be recovered to the saline control group by erinacine A and HEM (Figure 6A). Figure 6 shows that motor function of MPTP injection of erinacine A and HEM was also determined by a rotary rod test, in comparison to the MPTP group. After MPTP treatment, the mice showed a significant motor deficit, as suggested by a decrease in the times, as compared with the control group mice (*p* < 0.05, n = 6). The erinacine A and *H. erinaceus* mycelium groups showed reduced motor dysfunction in a dose-dependent manner compared with the MPTP group (*p* < 0.01, n = 6; Figure 6A). In addition, the post-treatment with erinacine A and HEM also attenuated dopaminergic neurons’ protein tyrosine hydroxylase in the substantia nigra and putamen from MPP^+^-induced neurotoxicity (Figure 6B). As shown in Figure 6B, quantification of the dopaminergic neurons’ cell protein tyrosine hydroxylase in the substantia nigra showed that the dosage of erinacine A and HEM administration increased the expression of protein tyrosine hydroxylase compared with the MPTP group (untreated MPTP = 5% ± 5%, * < 0.05; erinacine A treated MPTP = 75 ± 10, # < 0.05; HEM treated MPTP = 40 ± 10 and 80 ± 10, # < 0.05). Together, our data indicated that post-treatment with erinacine A could rescue the motor deficit and loss of the dopaminergic neurons’ protein tyrosine hydroxylase in an MPTP model.

### 3.5. Post-Treatment with Erinacine A Reversed the PAK1/AKT/LIMK2/ Pathways in MPTP-Treated Animals

In order to verify the effects of the oral administration of HEM and erinacine A on the relationships among neuroprotection, inflammation, and ER stress in the brain, the Fas protein was selected for further examination using immunohistochemistry assays in vivo after MPTP injury and HEM treatment.

To further determine if the possible mechanism of erinacine A through the i.p. and oral administrations of HEM decreased MPTP-induced neurotoxins and movement deficit due to the relationship between neuroprotection and neuronal signaling in the brain, we measured the protein level of PAK1/AKT/LIMK2 and p-ASK1 Thr845 in the brain striatum area after administering HEM by IHC analysis for further examination in vivo after MPTP injury and HEM treatment. Our data showed that the protein level of PAK1/AKT/LIMK2 decreased in the MPTP group. Interestingly, the post-treatment with erinacine A reversed the MPTP-decreased protein level of PAK1/AKT/LIMK2 to the level of the saline control group (Figure 7). In addition, the cell death markers p-ASK1 Thr845 and p21 were upregulated by the MPTP treatment, whereas the post-treatment with erinacine A and HEM administration reduced the MPTP-induced protein level of p-ASK1 Thr845 and p21 (Figure 8). Our data indicated that HEM and erinacine A not only activated the PAK1/AKT/LIMK2 survival pathways but also inhibited the cell death signals of p-ASK1 Thr845 and p21 in the brain striatum area in the MPTP group, while HEM post-treatment improved the motor deficit and rescued the neuronal loss in the MPTP-lesioned mice.

## 4. Discussion

Many studies have shown that reducing oxidative-stress-induced free radicals via an antioxidant effect can protect against neuronal damage and MPTP toxicant-induced degenerative diseases in response to chemopreventive drugs, such as dietary phytochemicals and mushrooms [33,34]. In addition, our previous study showed that *H. erinaceus* mycelium and extracted erinacine A could be used to investigate in vitro and in vivo anti-neuroinflammation involved in the protection agonist endoplasmic reticulum stress, which is induced by the loss of dopaminergic neurons and disordered motor function by MPTP injury [25]. The present study demonstrated that erinacine A with post-treatment regimens provides potent beneficial effects in MPTP models of dopaminergic degeneration. Our data revealed that post-treatment with erinacine A could modulate multiple signaling pathways involved in neuronal survival and cell death pathways in order to inhibit MMP^+^-induced progressive neuron death and the generation of ROS (Figure 2 and Figure 3). The signaling molecules affected by erinacine A included the survival factors PAK1, cdc42, AKT, LIMK2, ERK, and Cofilin, as well as the apoptotic regulators IRE1á, TRAF2, ASK1, GADD45, and p21 (Figure 4). In addition, we employed several neuron cell death assays to confirm that the MPP^+^ damage, as measured by brain neuron death, was significantly decreased with the erinacine A treatment at the concentration of 5 ìM for 24 h in the N2a cells (Figure 6). Using an in vivo MPTP animal model, our data demonstrated that the dopaminergic lesions and oxidative stress in the striatum, substantia nigra, and putamen, as well as the occurrence of motor disorder, were significantly decreased after treatment with HEM and its ethanol extraction of erinacine A (Figure 5, Figure 6, Figure 7 and Figure 8). Together, our data revealed that erinacine A treatment could not only increase the survival pathways but also reduce the apoptotic neurons induced by MPP^+^.

Consistent with our previous studies, pretreatment or post-treatment with *Hericium erinaceus* both displayed the antiapoptotic capability of DA neurons in vivo. In addition to its effects on central nervous system (CNS), erinacine A also exerts many biological properties [35], such as antioxidant activity, hypolipidemic activity, hemagglutinating activity, antimicrobial activity, antiaging activity, immune modulation, and anticancer activity. It will be interesting to know if erinacine A can directly cross the blood brain barrier (BBB) or first target the peripheral organs and affect neuronal functions in the brain.

Human brains are composed of neurons and non-neural cells. Although the anti-inflammatory property of erinacine A has been found [36], the impact of erinacine A on glia cells remains totally unknown. Regarding the study of this mechanism, treatment of erinacine A has been found to have potent effects on multiple signaling pathways, including anti-ROS-mediated TRAF2-IRE1-GADD45, the activated PAK1/AKT/LIMK2 survival pathways, and upregulation of the NGF protein [37]. However, potential receptors for the binding of erinacine A and how these signals interact with each other require further investigation.

Over the past decade, the development of drugs capable of treating disease progression in neurodegenerative diseases such as Parkinson’s disease has failed [38]. The lack of suitable Parkinson’s disease models for drug development has been considered one particular reason for the failure to translate these drugs into the clinic [34,38]. Studies have shown that herbal medicines or natural phytochemicals from certain plants have the capability to improve the motor or nonmotor symptoms in PD and delay the progression of neuroprotective effects in various experimental models of neurological disorders [39]. In addition to erinacine A, *Hericium erinaceus* extracts also contain other compounds similar to erinacines (B–I) and fruit bodies (hericenones C–H) [40]. Some reports have demonstrated that these components are the source of many bioactive and biological properties with low cell toxicity [41]. Determining the biological functions of each compound could result in a comprehensive view of the bioactive spectrum of *Hericium erinaceus* and its clinical application in the treatment of neurodegenerative disorders. Furthermore, reducing oxidative-stress-induced free radicals via an antioxidant effect has been shown to protect against the neuronal damage caused by neurotrophin deficiencies and toxin-induced degenerative diseases in response to chemopreventive drugs, such as dietary phytochemicals, phenols, alkaloids, flavonoids, and mushrooms. In addition to the MPTP model, there are two inflammation-based progressive neurodegenerative models, which consist of either a systemic injection of lipopolysaccharides (LPS) to either C57BL/6J [42] or transgenic mice overexpressing human A53T mutant α-synuclein [43]. Although erinacine A has shown potent neuroprotective effects in the MPTP model, we also plan to apply it to the two abovementioned inflammation-based progressive neurodegenerative models before beginning clinical trials.

## 5. Conclusions

The present study demonstrated that HEM and its ethanol extraction of erinacine A treatment exhibited preventive and therapeutic actions in the restoration of dopaminergic degeneration as well as motor dysfunctions in a dose-dependent manner. Our study firstly demonstrated that post-treatment with erinacine A could reduce MPTP-induced neurotoxicity via activation of the PAK1, AKT, LIMK2, MEK, and Cofilin survival pathways, as well as reduction of the IRE1α, TRAF2, ASK1, GADD45, and p21 cell death pathways in N2a neuron cells and in an MPTP model of Parkinson’s disease (Figure 9). Accordingly, erinacine A may be a potentially valuable neuroprotective and therapeutic agent that could be used to improve pathological conditions and behavioral deficits during PD treatment.

## Figures and Tables

**Figure 1 antioxidants-09-00137-f001:**
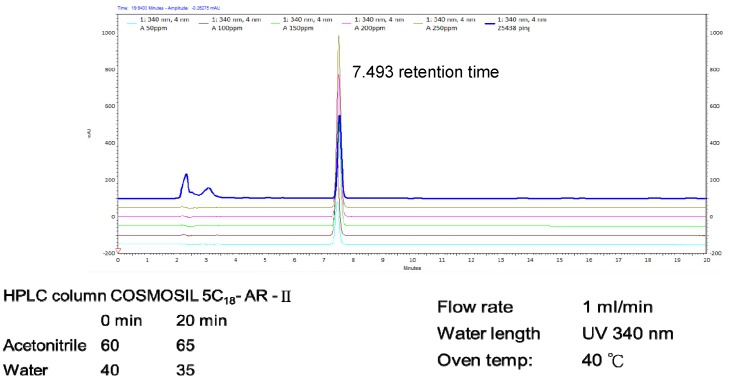
HPLC analysis and LC-MS analysis of the ethanol *Hericium erinaceus* mycelium (HEM) extract. The retention time peak at 7.493 min was erinacine A (UV detection at 340 nm).

**Figure 2 antioxidants-09-00137-f002:**
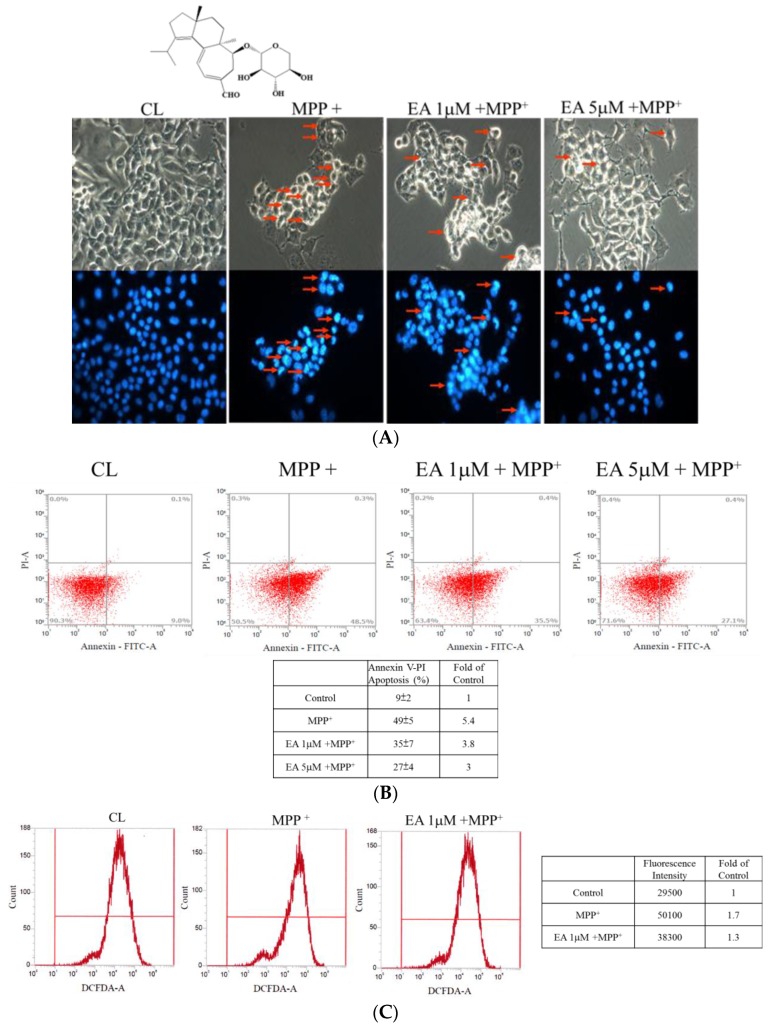
Erinacine A prevention of 1-methyl-4-phenylpyridinium (MPP^+^)-induced cell death and reactive oxygen species (ROS) generation of Neuro-2a (N2a) cells. (**A**) Cell apoptosis of the N2a cell line treated with 0.1% dimethyl sulfoxide (DMSO), MPP^+^, or MPP^+^ and erinacine A (1 and 5 µM) at 24 h measured by DAPI staining under fluorescence microscopy. The red arrows indicate the condensed apoptotic cells with fragmented nuclei at a magnification of ×200. (**B**) Flow cytometry analysis of the N2a cell line treated with 0.1% DMSO, MPP^+^, or MPP^+^ and erinacine A (1 and 5 µM) at 24 h stained by fluorescein isothiocyanate (FITC)-conjugated annexin V and propidium iodide (PI). The percentages of these treated cells with positive annexin V and/or PI and their changes (fold increase of untreated control) are shown in each quadrant. (**C**) After being incubated with H2DCFDA, the intracellular ROS of the N2a cell line treated with 0.1% DMSO, MPP^+^, or MPP^+^ and erinacine A at 24 h was detected by FACS analysis. Typical FACS profiles for H2DCFDA in the cells under different treatments are shown in the upper panel; the fluorescence intensity of each treatment and their changes (fold increase of untreated control) are shown in the bottom table.

**Figure 3 antioxidants-09-00137-f003:**
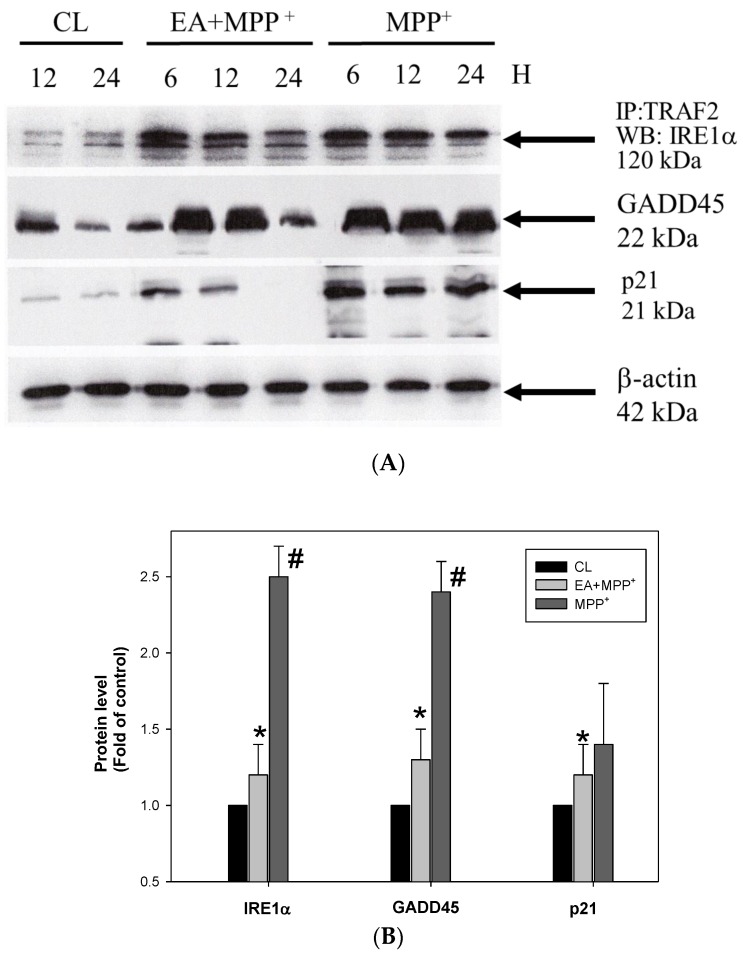
Erinacine A reduced MMP^+^ and increased the interaction of TRAF2 and IRE1, as well as expression of GADD45. (**A**) After the cells were treated with 0.1% DMSO, MPP^+^, or MPP^+^ and erinacine A for 24 h, immunoprecipitation and immunoblotting analyses were performed to detect the interaction of TRAF2 and IRE1 and the expression of GADD45, respectively. β-actin served as the internal control. (**B**) Quantification of the immunostaining for TRAF2, IRE1, and GADD45 at 24 h by densitometric analysis from three independent experiments; the data were normalized to that of β-actin and expressed as the fold increase of the untreated control. Two independent experiments were performed in duplicate; * *p* < 0.01, as compared to the control group; ^#^
*p* < 0.01, as compared to the MPP^+^-treated group.

**Figure 4 antioxidants-09-00137-f004:**
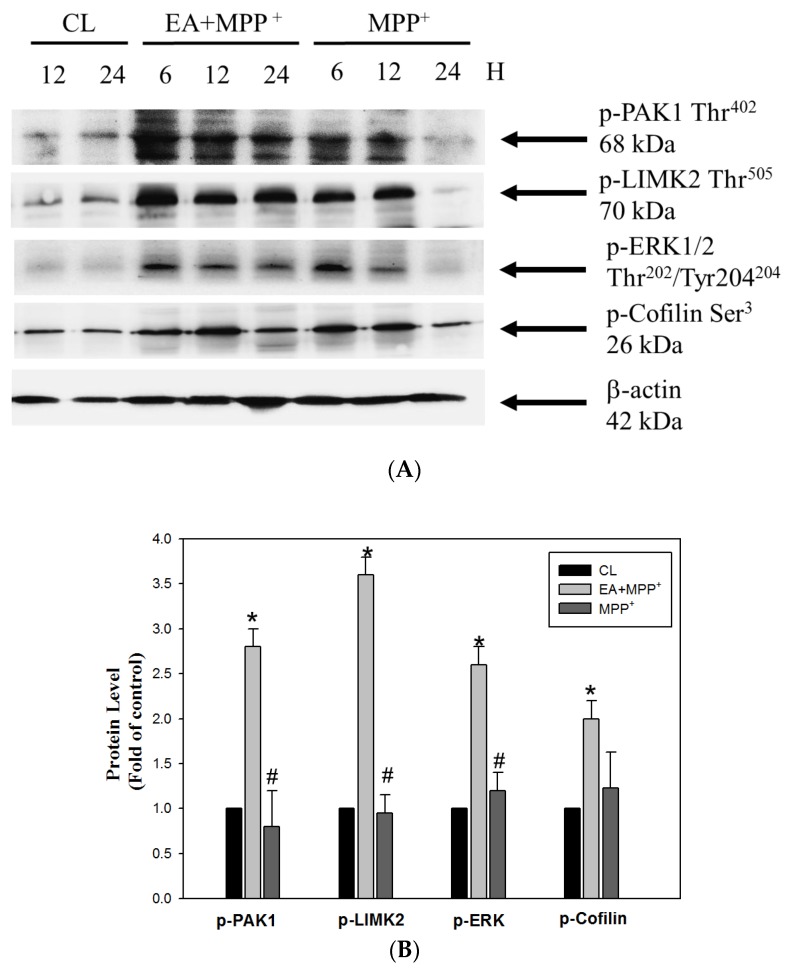
(**A**) Erinacine A enhanced phosphorylation of PAK1/LIMK2/MEK/Cofilin under MMP^+^ challenge. Western blotting for phosphorylation of PAK1, LIMK2, MEK, and Cofilin, as well as β-actin (loading control) in the N2a cells treated with 0.1% DMSO, MPP^+^, or MPP^+^ and erinacine A for 24 h. (**B**) The quantitative procedure of the immunoblots at 24 h was described previously. Data are expressed as the fold increase of the control group (mean ± SEM); * *p* < 0.01, as compared to the control group; ^#^
*p* < 0.01, as compared to the MPP^+^-treated group.

**Figure 5 antioxidants-09-00137-f005:**
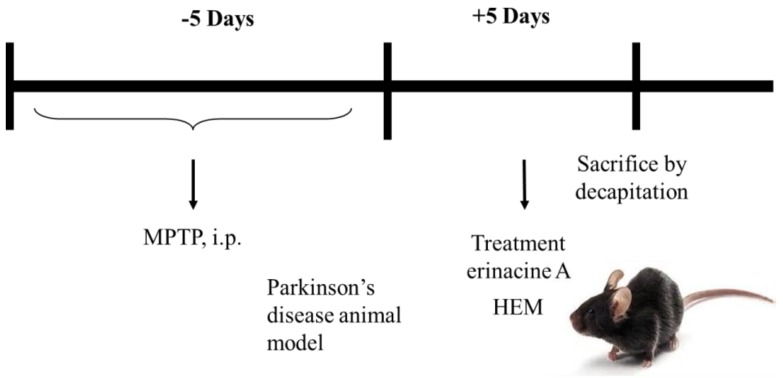
Simplistic flow chart of the therapeutic activities of HEM in a 1-methyl-4-phenyl-1,2,3,6-tetrahydropyridine (MPTP)-treated animal model.

**Figure 6 antioxidants-09-00137-f006:**
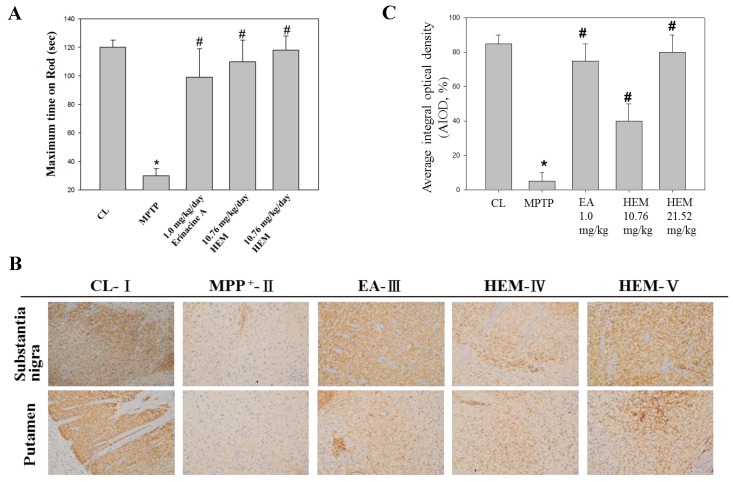
Post-treatment with erinacine A weakened the MPTP-induced neurotoxin and movement deficit. (**A**) Erinacine A intraperitoneally postadministered in the MPTP-treated animals after 5 days. An animal behavior test was performed 5 days after erinacine A was administered and then the animals were scarified by decapitation. Accelerating rotarod tests for the animals with MPP^+^ or MPP^+^ and erinacine A were performed as indicated after 1 day of training. Quantified data for the elapsed time during the six test trials is shown. (**B**) Immunohistochemistry (IHC) staining for the global IHC intensity of all dopaminergic neurons’ protein tyrosine hydroxylase in the substantia nigra of the animals. (**C**) Quantitative immunohistochemical protein tyrosine hydroxylase was evaluated according to the average integrated optical density (AIOD). The positive stained area was evaluated from three randomly selected observation fields for each brain section. Data are expressed as mean ± S.D. (n = 6/group); * *p* < 0.05, as compared to the control group, magnification ×400; * *p* < 0.01, as compared to the control group; # *p* < 0.01, as compared to the MPP^+^-treated group.

**Figure 7 antioxidants-09-00137-f007:**
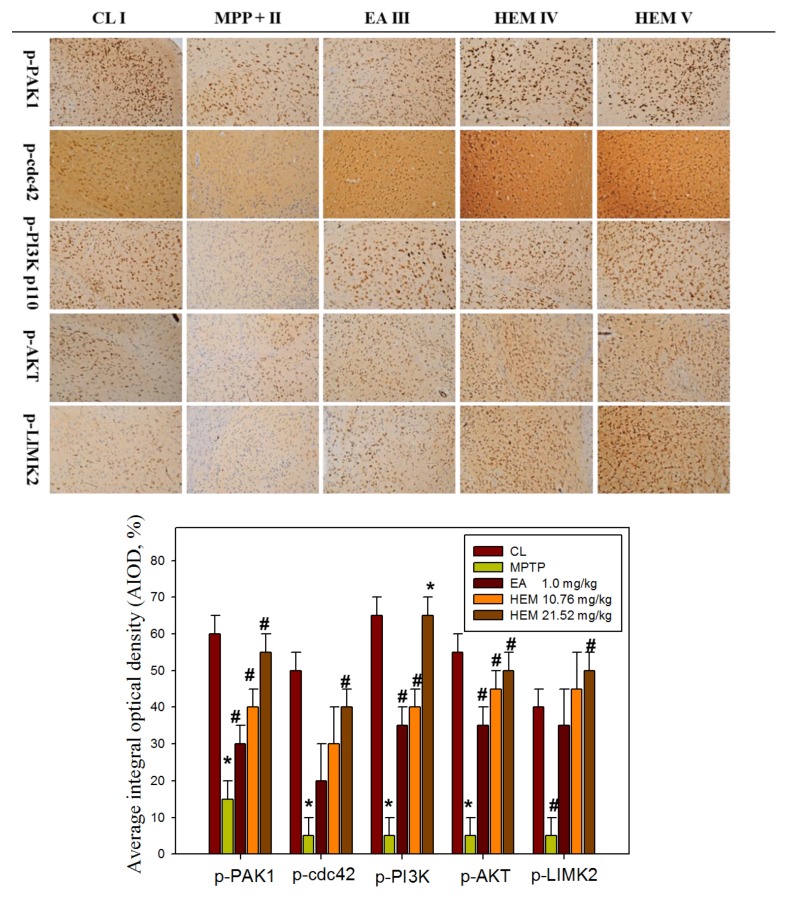
Post-treatment with erinacine A reversed the PAK1/AKT/LIMK2/ pathways in the MPTP-treated animals. Representative IHC images of p-PAK1, p-cdc42, p-PI3K p110, p-AKT, and p-LIMK2 in the brain region of the animals with MPP^+^ or MPP^+^ and erinacine A are as indicated. Quantified data of the average integral optical density are expressed as the mean ± SEM (fold increase of untreated animals). Data are expressed as mean ± S.D. (n = 6/group); * *p* < 0.01, as compared to the control group; # *p* < 0.01, as compared to the MPP^+^-treated group.

**Figure 8 antioxidants-09-00137-f008:**
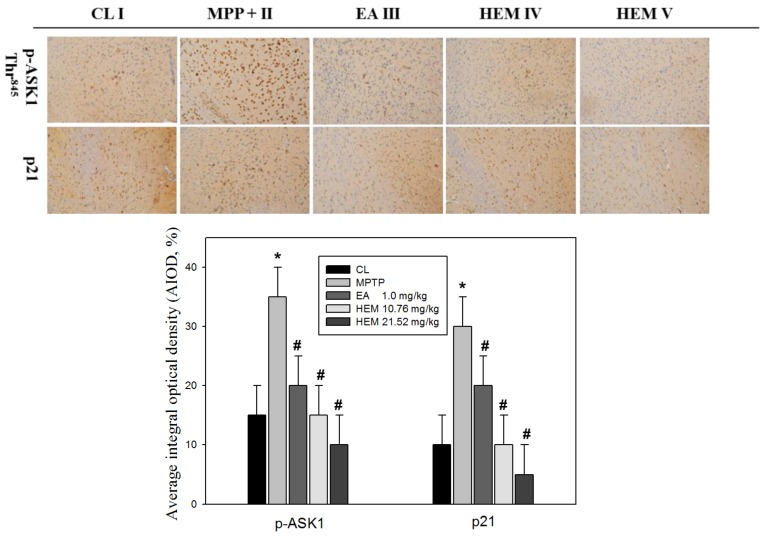
Post-treatment with erinacine A reduced the p-ASK1 and p21 expression in MPTP-treated animals. Representative IHC images of p-ASK1 and p21 in the brain region of the animals with MPP^+^ or MPP^+^ and erinacine A are as indicated. Quantified data of the average integral optical density are expressed as the mean ± SEM (* *p* < 0.01, as compared to the control group; # *p* < 0.01, as compared to the MPP^+^-treated group). Data are expressed as mean ± S.D. (n = 6/group).

**Figure 9 antioxidants-09-00137-f009:**
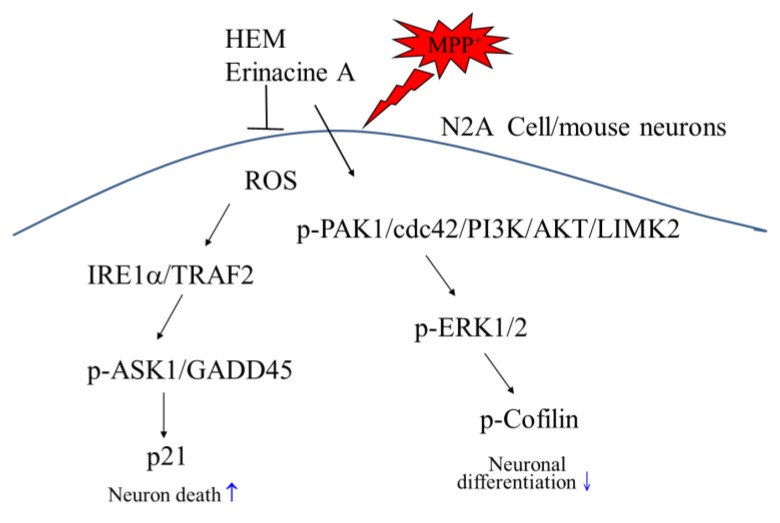
Schematic presentation of erinacine A prevention against MPP^+^-induced neurotoxicity via either induction of the p-PAK1/cdc42/PI3K/AKT/LIMK2 cell survival pathways or the reduction of the IRE1α/TRAF2/ASK1/GADD45/p21 cell death pathway. In addition to decreasing the cell death pathways (IRE1α/TRAF2, ASK1, GADD45, and p21), our findings revealed a new paradigm for erinacine A to enhance the neuronal survival pathways (PAK1/AKT/LIMK2/ERK/Cofilin) that could protect neurons against MPP^+^ injury.

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
