# Peer review of "Post-Treatment with Erinacine A, a Derived Diterpenoid of H. erinaceus, Attenuates Neurotoxicity in MPTP Model of Parkinson’s Disease"

_antioxidants, 2020, doi:10.3390/antiox9020137_

Round 1
Reviewer 1 Report
In the present manuscript Lee et al. explored the neurorestorative potential of erinacine A, an active compound derived from the edible mushroom Hericium erinaceus, using MPP+/MPTP neurotoxic models of Parkinson’s disease both in vivo and in vitro. The authors reported that post-treatment with erinacine A decreased ROS formation in MPP+-treated mouse neuroblastoma N2a cells and thus promoted cell survival by 1) reducing the interaction between TRAF2 and IRE1 and the subsequent activation of the unfolded protein response (UPR); and 2) modulating the pro-survival pathways PAK1/LIMK2/MEK/Cofilin. In vivo, treatment with erinacine A was able to reverse the neurotoxic effects of MPTP in a subacute mouse model, recovering nigral dopaminergic function and ameliorating motor behaviour in a dose-dependent manner by modulation of the same pathways previously described in cell culture.
Although the study is potentially interesting and the models used relevant for the disease, a number of concerns arise when reading the manuscript, including issues about data analysis and novelty. In addition, a large number of mistakes in figures and references, as well as numerous errors in English grammar do not favour an easy and comprehensive reading.
Specific comments are listed below.
MAJOR
1. The key finding of the work of Lee et al. was the neurorestorative effect of erinacine A in an MPTP mouse model. However, the method used by the authors to asses neurodegeneration in these animals is not reliable. The precise methodology used is not very clear but it seems that they have evaluated optical density in randomly selected microscopy fields at high magnification (100x, 200x, compared to the 40x normally used in stereology), which would highly depend on individual variation and any type of IHC artefacts.
The golden standard for dopaminergic neurodegeneration assessment is the estimation of the absolute number of TH+ neurons in the substantia nigra by sterological cell counting with a specific stereology setup such as Stereoinvestigator. Stereology is performed in a representative number of sections covering the whole substantia nigra (sampling every 6-10 sections) to make a stack of optical sections that allow scaling to the total span of the nigra. This method is unbiased and not influenced by differences in TH immunostaining. If stereology is not possible, an alternative method may be counting or measuring global IHC intensity of all TH+ cells in all the nigral sections from each animal.
Furthermore, the methodology used by the authors does not account for TH downregulation, a phenomenon that has been demonstrated to precede dopaminergic cell death and with a high relevance when testing neurorestorative drugs. Thus, it would be crucial towards further clinical application to understand whether the effect of erinacine A might be solely due to a recovery of the TH+ phenotype in dysfunctional (but not dead) dopaminergic neurons.
2. Whereas it is true that the authors report for the first time the neurorestorative (post-MPTP treatment) potential of erinacine A in cells and in vivo, the work presented here follows a publication from 2016 (Kuo et al., J Transl Med. 2016) in which pre-treatment with erinacine A had neuroprotective effects in the same MPTP mouse model. Neuroprotection effect of erinacine A was also mediated through the modulation of the unfolded protein response (UPR), just as the neurorestorative effect shown here. As it stands, it is not clear to me that the present results constitute a substantial novelty, except for the involvement of the PAK1/LIMK2//MEK/Cofilin pathway. In addition, phosphorylation of these proteins seems to be increased by MPP+ treatment in cell culture, whereas treatment with erinacine A reduced phosphorylation back to basal levels. In vivo data, however, shows completely the opposite. This difference should be mentioned and explained in the discussion section.
Antioxidants with a proved efficacy in animal models usually fail or show modest effects in human clinical trials. The MPTP model, although reproducing many parkinsonian features, arises from a very specific and acute insult, which is very different from the complex origin and slow clinical progression in Parkinson’s disease. Demonstration of the efficacy of erinacine A in a different disease model would strength the conclusions of the study.
3. The text is plenty of avoidable mistakes in references, figure legends and English grammar, suggesting a not very careful preparation of the manuscript that discourages reading.
MINOR
1. Title (line 4): Parkinson’s disease should be written in capital letters
2. Introduction (line 70): it would be appropriate that the authors briefly introduce the molecular mechanisms involved in MPTP neurotoxicity (at least how MPTP increases ROS production), especially when antioxidants are being tested. Also, some reference to the general phenotype of the MPTP model used here (subacute or chronic regime) would be appreciated.
3. Introduction: literature is not properly cited in some cases, with references mixed, missing or not related to the assertions done in the text. Please check carefully the whole introduction section, especially Ref. 1, 5, 6 and 16.
4. Results (line 210) and Fig. 1B. The interpretation of the cell death/viability results is not clear at all, nor is the quantification of apoptosis and cell survival (a 3% increase in neuron survival is really significant?). It would be recommendable to rewrite this paragraph in a more clear way, adding to the flow cytometry graphs in Fig. 1B a bar graph that shows cell death/viability quantification, together with their corresponding error bars and significance.
5. Fig 1C: Despite DCFH-DA has been widely used to measure intracellular ROS, its non-specific chemistry can cause false positives and artefacts that make interpretation of the results difficult. In addition of not being a direct measure of ROS, DCFH-DA can leach out of the cells, so it cannot be considered an actual intracellular marker. Taking all this into account, the general recommendation is to avoid DCFH-DA if possible, or at least to complement it with other techniques for measuring intracellular ROS.
Instead of the fluorescence intensity table, a bar histogram including error bars and statistical significance would be helpful.
6. Fig. 2: Figure legend is not clear and should be rewritten. One does not know for sure if the graph corresponds to the WB or to the IF, or what is exactly the number of replicates in each experiment. First, upper panel should be named A) and lower panel B) to make the figure legend easier to follow. Next, both A and B should include on one side the representative image of the WB or IF respectively and on the other side the corresponding graph for signal quantification with error bars and significance.
Figure legend mentions TRAF2, IRE1 and GADD45, but p21 is totally missing.
p21 quantification in the lower graph indicates significant differences, which seems unlikely, given the error bars. Please recheck and include the type of statistical test and post-hoc test performed here (this point applies to all the figures in the manuscript).
7. Fig. 3: The effect of MPP+ on phosphorylation of PAK1, LIMK2 and ERK seems to be transient, since at 24 h phosphorylation levels are similar to controls. However, quantification after 24 h shows high phosphorylation levels of these proteins in the MPP+ group. Is this a mistake or is there any other WB that could be more representative of the data? Authors may like to discuss this transient effect in the discussion section.
8. Flow chart (Fig 5A) is actually not in this figure as it stands in the figure legend, but as a separate Fig 4. Thus B, and C, do neither correspond to what is shown in the figure.
In the images from the TH+ immunohistochemistry in Fig 5B (5C according to the legend), the substantia nigra is not recognizable, even less can the differences in cell death be distinguished. Dopaminergic quantification should be represented in absolute numbers, and not as a percentage of the control.
9. Fig.7: figure legend corresponds to Fig. 6 and not to what is shown in Fig 7. Please, enter the right legend.
10. Statistical analysis is not well described. According to the methods section, all analyses have been done using one-way ANOVA, but there is no indication in the methods or in the figures legends of the post-hoc analysis used or to the specific number of samples in each individual experiment.
Author Response
RE: Revised version of antioxidants-650790
Dear Megan Xu
Assistant Editor,
Enclosed please find one revised version entitled: Post-treatment with erinacine A, a derived diterpenoid of H. erinaceus attenuates neurotoxicity in MPTP model of parkinson's disease, which we would like to submit for publication in Antioxidants.
This revised version has been carefully corrected according editor and referee’s reports point-by-point. We appreciate these valuable comments to strengthen our presentation. Please inform me if any revision is needed. The file marked change in blue color.
Furthermore, I would verify that no part of the manuscript is under consideration for publication elsewhere and it will not submit elsewhere if accepted by Antioxidants and not before the Editorial Office has reached a decision.
Sincerely yours,
Hsing-Chun Kuo, Ph.D.
Associate Professor,
Institute of Nursing and Department of Nursing,
Chang Gung University of Science and Technology,
Chia-Yi Campus, Taiwan.
E-mail: guscsi@gmail.com
TEL: +886-5-3628800
FAX: +886-5-3628866
Response to Reviewer 's Comments
Reviewer(s)' Comments to Author:
Reviewer #1:
Comments and Suggestions for Authors
Major:
The key finding of the work of Lee et al. was the neurorestorative effect of erinacine A in an MPTP mouse model. However, the method used by the authors to asses neurodegeneration in these animals is not reliable. The precise methodology used is not very clear but it seems that they have evaluated optical density in randomly selected microscopy fields at high magnification (100x, 200x, compared to the 40x normally used in stereology), which would highly depend on individual variation and any type of IHC artefacts.
Response: We appreciate this remark of comment and the methodology was revised in manuscript.
The digital images were captured and quantitatively evaluated from three randomly selected microscopic fields (200x magnification) on each slide to measuring global IHC intensity of all dopaminergic neurons protein tyrosine hydroxylase in all the nigral sections from each animal, based on the average integral optical density (AIOD = positive area×optical density /total area). As shown in Fig. 5B, quantification of the dopaminergic neuron cells protein tyrosine hydroxylase in substantia nigra showed that the dosage of erinacine A and HEM administration were contented to increase the expression of protein tyrosine hydroxylase compared to the MPTP group (untreated MPTP = 5% ± 5, * <0.05; erinacine A treated MPTP = 75 ± 10, # <0.05; HEM treated MPTP = 40 ± 10 and 80 ± 10, # <0.05). Together, our data indicated that post-treatment of Erinacine A can rescue the motor deficit and loss of dopaminergic neuons protein tyrosine hydroxylase in a MPTP model.
Whereas it is true that the authors report for the first time the neurorestorative (post-MPTP treatment) potential of erinacine A in cells and in vivo, the work presented here follows a publication from 2016 (Kuo et al., J Transl Med. 2016) in which pre-treatment with erinacine A had neuroprotective effects in the same MPTP mouse model. Neuroprotection effect of erinacine A was also mediated through the modulation of the unfolded protein response (UPR), just as the neurorestorative effect shown here. As it stands, it is not clear to me that the present results constitute a substantial novelty, except for the involvement of the PAK1/LIMK2//MEK/Cofilin pathway. In addition, phosphorylation of these proteins seems to be increased by MPP+ treatment in cell culture, whereas treatment with erinacine A reduced phosphorylation back to basal levels. In vivo data, however, shows completely the opposite. This difference should be mentioned and explained in the discussion section.
Response: We appreciate this remark of reviewer #1’s comment and respond to the comment as follows: From previous studies, Hericium erinaceus mycelium and erinacine A have neuroprotective potential through blocking oxidative stress MPTP-induced neuron damage on the ER stress triggering IRE1a/TRAF2 complex formation, which inhibits an apoptosis cascade. These study firstly demonstrate that post-treatment of erinacine A, as a therapeutic agent, reduces MPTP-induced neurotoxicity via activation of the PAK1,AKT, LIMK2, MEK and Cofilin survival pathways and reduction of the IRE1α, TRAF2, ASK1, GADD45 and p21 cell death pathways in N2a neuron cells and in MPTP model of parkinson's disease. In N2a cells, the result showed phosphorylation of these proteins seems to be induced by MPTP+ treatment in cell culture because of feedback mechanism. Quantitative procedure of immune blots phosphorylation of PAK1/LIMK2/MEK/Cofilin at 24 hrs was determined by densitometric analysis. These data suggested that phosphorylation of these proteins seems to be decreased by MPP+ treatment in cell culture at 24 hrs.
Antioxidants with a proved efficacy in animal models usually fail or show modest effects in human clinical trials. The MPTP model, although reproducing many parkinsonian features, arises from a very specific and acute insult, which is very different from the complex origin and slow clinical progression in Parkinson’s disease. Demonstration of the efficacy of erinacine A in a different disease model would strength the conclusions of the study.
The text is plenty of avoidable mistakes in references, figure legends and English grammar, suggesting a not very careful preparation of the manuscript that discourages reading.
Response: Thanks for the comments. These are described in page 16, line 409 to line 414 in the text. The error in manuscript were revised in blue color.
In addition to MPTP model, there are two inflammation-based progressive neurodegenerative models in which of a systemic injection of LPS to either C57BL/6J [42] or transgenic mice over-expressing human A53T mutant α-synuclein [43]. Although Erinacine A has shown potent neuroprotective effects in MPTP model, we also plan to apply it into these two inflammation-based progressive neurodegenerative models before entrance of clinical trial.
Minor
Title (line 4): Parkinson’s disease should be written in capital letters
Response: Thanks for the comments. This is revised in page 1, line 4.
2.Introduction (line 70): it would be appropriate that the authors briefly introduce the molecular mechanisms involved in MPTP neurotoxicity (at least how MPTP increases ROS production), especially when antioxidants are being tested. Also, some reference to the general phenotype of the MPTP model used here (subacute or chronic regime) would be appreciated.
Response: Thanks for the comments. These are described in page 2, line 64 to line 75 in the text.
One of the well-established PD’s animal model is implantable MPTP (1-methyl-4-phenyl-1,2,3,6-tetrahydropyridine) into animals in order to produce the neurotoxin MPP+(1-methyl-4-phenylpyridinium), which causes permanent symptoms of Parkinson's disease by destroying dopaminergic neurons in the substantia nigra of the brain [4], similar to those seen in PD. Basically, the initial biochemical event of neurotoxic action of MPTP is a two-step oxidation by monoamine oxidase B in glial cells to MPP+ [5]. Following uptake by the synaptic dopamine reuptake system, MPP+ are further concentrated by the electrical gradient of the inner membrane and then more slowly penetrate the hydrophobic reaction site on NADH dehydrogenase. MPP+ combined with NADH dehydrogenase leads to cessation of oxidative phosphorylation, ATP depletion, and cell death. On the other hand, 1-methyl-4-phenylpyridinium (MPP+) treatment generates reactive oxygen species in dopaminergic neurons via an NADPH oxidase-dependent two-wave cascade [6].
Introduction: literature is not properly cited in some cases, with references mixed, missing or not related to the assertions done in the text. Please check carefully the whole introduction section, especially Ref. 1, 5, 6 and 16.
Response: Thanks for the comments. These references are revised in Ref. 5, 6, 7, 8 and 19.
Results (line 210) and Fig. 1B. The interpretation of the cell death/viability results is not clear at all, nor is the quantification of apoptosis and cell survival (a 3% increase in neuron survival is really significant?). It would be recommendable to rewrite this paragraph in a more clear way, adding to the flow cytometry graphs in Fig. 1B a bar graph that shows cell death/viability quantification, together with their corresponding error bars and significance.
Response: Thanks for the comments. We appreciate this remark of the Editor’s comment. we have revised Western blots and provide better quality pictures cell death/viability quantification and a bar histogram, together with their corresponding error bars in the revised version of the Fig 2B. It has revised in the manuscript, page 8, lines 233 to 237.
Fig 1C: Despite DCFH-DA has been widely used to measure intracellular ROS, its non-specific chemistry can cause false positives and artefacts that make interpretation of the results difficult. In addition of not being a direct measure of ROS, DCFH-DA can leach out of the cells, so it cannot be considered an actual intracellular marker. Taking all this into account, the general recommendation is to avoid DCFH-DA if possible, or at least to complement it with other techniques for measuring intracellular ROS. Instead of the fluorescence intensity table, a bar histogram including error bars and statistical significance would be helpful
Response: We appreciate this remark of the Editor’s comment. In addition to addressing the comment specifically, we have revised DCFH-DA and a bar histogram and provide better quality pictures in the revised version of the Fig 2C.
Fig. 2: Figure legend is not clear and should be rewritten. One does not know for sure if the graph corresponds to the WB or to the IF, or what is exactly the number of replicates in each experiment. First, upper panel should be named A) and lower panel B) to make the figure legend easier to follow. Next, both A and B should include on one side the representative image of the WB or IF respectively and on the other side the corresponding graph for signal quantification with error bars and significance. Figure legend mentions TRAF2, IRE1 and GADD45, but p21 is totally missing. p21 quantification in the lower graph indicates significant differences, which seems unlikely, given the error bars. Please recheck and include the type of statistical test and post-hoc test performed here (this point applies to all the figures in the manuscript)..
Response:
We appreciate this comment and the quality of Fig 3A WB and p21 quantification have considerably improved. To accommodate the comment, we have added to the revised version of statistical test here in Fig 3B.
Fig. 3: The effect of MPP+ on phosphorylation of PAK1, LIMK2 and ERK seems to be transient, since at 24 h phosphorylation levels are similar to controls. However, quantification after 24 h shows high phosphorylation levels of these proteins in the MPP+ group. Is this a mistake or is there any other WB that could be more representative of the data? Authors may like to discuss this transient effect in the discussion section.
Response:
We appreciate this comment and the quality of Fig 4A WB has considerably improved. To accommodate the comment, we have added to the revised version of statistical test here in Fig 4B at 24 h. The result showed MPP+ on phosphorylation of PAK1, LIMK2 and ERK seems to be transient, in addition, phosphorylation levels are similar to control group at 24 h.
Flow chart (Fig 5A) is actually not in this figure as it stands in the figure legend, but as a separate Fig 4. Thus B, and C, do neither correspond to what is shown in the figure. In the images from the TH+ immunohistochemistry in Fig 5B (5C according to the legend), the substantia nigrais not recognizable, even less can the differences in cell death be distinguished. Dopaminergic quantification should be represented in absolute numbers, and not as a percentage of the control.
Response:
We appreciate this comment and we have added to the revised version of the Fig 6 legend and Fig 6 B, 6C.
The method for measuring global IHC intensity of all TH+ cells in all the nigral sections from each animal. As shown in Fig. 6B, quantification of the dopaminergic neuron cells protein tyrosine hydroxylase in substantia nigra showed that the dosage of erinacine A and HEM administration were contented to increase the expression of protein tyrosine hydroxylase compared to the MPTP group (untreated MPTP = 5% ± 5, * <0.05; erinacine A treated MPTP = 75 ± 10, # <0.05; HEM treated MPTP = 40 ± 10 and 80 ± 10, # <0.05). These are described in the text from page 11 to line 297 to line 305.
Fig.7: figure legend corresponds to Fig. 6 and not to what is shown in Fig 7. Please, enter the right legend.
Response:
We appreciate this comment and we have added to the revised version of the Fig 8 legend from page 14 to line 370 to age 15 to line 375.
Statistical analysis is not well described. According to the methods section, all analyses have been done using one-way ANOVA, but there is no indication in the methods or in the figures legends of the post-hoc analysis used or to the specific number of samples in each individual experiment.
Response:
We appreciate this comment and we have added to the revised version of the Fig legends of the post-hoc analysisn each individual experiment.
Data were expressed as mean ± S.D. (n = 6/group) of three independent experiments and were analysed by one-way ANOVA. The data were analysed using the SAS software statistical package ‘SigmaPlot’, version 9.0 (SAS Institute Inc., Cary, NC, USA).

Reviewer 2 Report
The manuscript "Post-treatment with erinacine A, a derived diterpenoid of H. erinaceus attenuates neurotoxicity in MPTP model of parkinson's disease"
is one of the good articles in an endeavor to get insights into the mechanism for Parkinson's disease and identify the therapeutic intervention.
In this article, Lee et al. determined that erinacine A treatment could restore dopaminergic neuronal loss and motor dysfunction.
The authors have found that Post-treatment of erinacine A suppresses MPTP-induced cytotoxicity of neuronal cells and hence ROS production via phosphorylation of the PAK1, AKT, LIMK2, ERK and Cofilin promoting survival pathway. Similarly, both in-vivo and in-vitro studies showed a reduction of cell death pathway that involves
IRE1alfa/TRAF2, p21, ASK1, and GADD45.
The strength of the article is that the author/s identified that eranicine could potentially neuroprotective and therapeutic agent for ameliorating pathological and behavior deficits in their model.
However, there are several other weaknesses in the article are as below.
-Line 3, 28: Hericium erinaceus should be italicized here and everywhere in the manuscript.
-Line 28-50: There are several short forms that are introduced in the abstract such as V-FITC, DCFDA, MMP, etc. Any abbreviations that are introduced for the first time should have full form text followed by short-form in the parentheses throughout the text.
In addition, you have "ROS" mentioned in lines 34, but you provided the full form in parentheses for ROS in lines 42. I suggest to define it in lines 34 and remove the full form in lines 42. Such pattern should be followed throught the manuscript.
-Line 43: Make sure "IRE1alfa-TRAF2" should contain hyphen or comma. See lines 74 and 381 to find out inconsistency "IRE1alfa/TRAF2" and"IRE1alfa, TRAF2" respectively.
-Line -65 or -84 or elsewhere: I suggest to write superscript for charge "+" in "MMP+"
-Line 107: EtOAc, HPLC short form should be expanded for the full form if it has been introduced in the text for the first time.
-Line 112-113: Figure 1 Make sure that "+" for "MMP+" is consistently written in the figure as well as mentioned earlier.
-In addition, Figure 1B, The font size and text should clear and consistent throughout the paper. Maybe post-processing for final figures is needed in this context.
-Line 174: Bold letter "apoptosis assay" should be removed
Line 176:"2X104 cells/ml" should be rechecked. I believe you want to have superscript for "4".
-Page 8, Line 226-227: GADD45 traces look not clean!
-Page 9, Line 245-246 Figure 3: Western blotting traces looks not clean, any better traces?
In addition, the quantification bar diagram in Figure 3 consists of labeling for "EA+MPP+" and "MPP+" specifying bar's legend (top right) which looks much similar. Please use different patterns/colors for these two or be consistent in Figure 2.
Line 152: "then then", please remove one "then"
Ling 153: please mention the ethanol concentration to be exact.
Line 155-159: It is not clear how long the primary antibody and secondary antibody were incubated and at what temperature?
Line 277-278: Please be consistent with "CL". Figure 5A has a "Control group (CL)", while earlier figures don't have full forms but only "CL". In Figure 5B, you have "CL I".
You need to define CL clearly in your text when you used for the first time, and you may use short-form always.
Line 306-308:Figure6: There are several groups, I suggest you provide a pattern inside the bar graph or use a different color.
For example "EA and HEM 21.52mg/kg" is difficult to distinguish. Similar goes to MPTP and HEM 10.76. Some "#" are attached to the error bar line. please separate it.
In the discussion, the author/s need to highlight the limitation of this study or future prospectus.
Line 375-383: Conclusion section should be consistent with the abstract section (See line 46-48). For example, in the abstract section (line 46-48),
the author saw dose-dependent changes, which should be mentioned in the conclusion section as well.
Line 383-390: In your schematic presentation, I suggest to align the arrow mark giving final look. For example, the arrow from ROS to IRE1α/TRAF2 can be moved slightly towards left.
An unnecessary bold letter or underline or color can be removed.
Author Response
RE: Revised version of antioxidants-650790
Dear Megan Xu
Assistant Editor,
Enclosed please find one revised version entitled: Post-treatment with erinacine A, a derived diterpenoid of H. erinaceus attenuates neurotoxicity in MPTP model of parkinson's disease, which we would like to submit for publication in Antioxidants.
This revised version has been carefully corrected according editor and referee’s reports point-by-point. We appreciate these valuable comments to strengthen our presentation. Please inform me if any revision is needed. The file marked change in blue color.
Furthermore, I would verify that no part of the manuscript is under consideration for publication elsewhere and it will not submit elsewhere if accepted by Antioxidants and not before the Editorial Office has reached a decision.
Sincerely yours,
Hsing-Chun Kuo, Ph.D.
Associate Professor,
Institute of Nursing and Department of Nursing,
Chang Gung University of Science and Technology,
Chia-Yi Campus, Taiwan.
E-mail: guscsi@gmail.com
TEL: +886-5-3628800
FAX: +886-5-3628866
Response to Reviewer 's Comments
Reviewer(s)' Comments to Author:
Reviewer #2:
Comments and Suggestions for Authors
Major:
-Line 3, 28: Hericium erinaceus should be italicized here and everywhere in the manuscript.
Response: Hericium erinaceus should be italicized in the manuscript.
-Line 28-50: There are several short forms that are introduced in the abstract such as V-FITC, DCFDA, MMP, etc. Any abbreviations that are introduced for the first time should have full form text followed by short-form in the parentheses throughout the text.
In addition, you have "ROS" mentioned in lines 34, but you provided the full form in parentheses for ROS in lines 42. I suggest to define it in lines 34 and remove the full form in lines 42. Such pattern should be followed throught the manuscript.
Response: We appreciate this comment and we have added to the revised version of the short-form in the manuscript.
-Line 43: Make sure "IRE1alfa-TRAF2" should contain hyphen or comma. See lines 74 and 381 to find out inconsistency "IRE1alfa/TRAF2" and"IRE1alfa, TRAF2" respectively.
-Line -65 or -84 or elsewhere: I suggest to write superscript for charge "+" in "MMP+"
-Line 107: EtOAc, HPLC short form should be expanded for the full form if it has been introduced in the text for the first time.
-Line 112-113: Figure 1 Make sure that "+" for "MMP+" is consistently written in the figure as well as mentioned earlier.
-In addition, Figure 1B, The font size and text should clear and consistent throughout the paper. Maybe post-processing for final figures is needed in this context.
Response: We appreciate this comment and the errors in manuscript were revised in blue color, IRE1alfa/TRAF2. We have revised to write superscript for charge "+" in "MMP+". Ethyl acetate (EtOAc) and high performance liquid chromatography (HPLC)were shown. We have revised better quality pictures in the revised version of the Fig 2B.
-Line 174: Bold letter "apoptosis assay" should be removed
Line 176:"2X104 cells/ml" should be rechecked. I believe you want to have superscript for "4".
-Page 8, Line 226-227: GADD45 traces look not clean!
-Page 9, Line 245-246 Figure 3: Western blotting traces looks not clean, any better traces?
In addition, the quantification bar diagram in Figure 3 consists of labeling for "EA+MPP+" and "MPP+" specifying bar's legend (top right) which looks much similar. Please use different patterns/colors for these two or be consistent in Figure 2.
Response: We appreciate this comment and the errors in manuscript were revised in blue color, Line 174: Bold letter "apoptosis assay" should be revised. We have revised better quality pictures in the revised version of the Fig 2B. The symbol 2 x 104 cells/ml was revised.
In addition to addressing the comment specifically, we have revised Western blots and provide better quality pictures in the revised version of the Figure 3 and Figure 4. Data of Figure 3 and Figure 4 presented in the Western blot are derived from a representative study, and relative quantization are calculated from the duplicate experiments and have considerably improved.
Line 152: "then then", please remove one "then"
Ling 153: please mention the ethanol concentration to be exact.
Line 155-159: It is not clear how long the primary antibody and secondary antibody were incubated and at what temperature?
Line 277-278: Please be consistent with "CL". Figure 5A has a "Control group (CL)", while earlier figures don't have full forms but only "CL". In Figure 5B, you have "CL I".
You need to define CL clearly in your text when you used for the first time, and you may use short-form always.
Line 306-308:Figure6: There are several groups, I suggest you provide a pattern inside the bar graph or use a different color.
For example "EA and HEM 21.52mg/kg" is difficult to distinguish. Similar goes to MPTP and HEM 10.76. Some "#" are attached to the error bar line. please separate it.
Response: We appreciate this comment and the errors in manuscript were revised in blue color, Line 152: Bold letter "apoptosis assay" should be revised. We have revised better quality pictures in the revised version of the Fig 2B. The symbol 2 x 104 cells/ml was revised.
Line 153: The sections were dehydrated using graded ethanol (10 %, 30 %, 50 %, 70% and 100 %).
The primary antibody and secondary antibody were incubated for 1h at 25℃.
We have revised better quality pictures in the revised version of the Fig 6B and 6B as well as Fig 7.
In the discussion, the author/s need to highlight the limitation of this study or future prospectus.
Response: Thanks for the comments. These future prospectus are described in page 16, line 409 to line 414 in the text. The error in manuscript were revised in blue color.
In addition to MPTP model, there are two inflammation-based progressive neurodegenerative models in which of a systemic injection of LPS to either C57BL/6J [42] or transgenic mice over-expressing human A53T mutant α-synuclein [43]. Although Erinacine A has shown potent neuroprotective effects in MPTP model, we also plan to apply it into these two inflammation-based progressive neurodegenerative models before entrance of clinical trial.
Line 375-383: Conclusion section should be consistent with the abstract section (See line 46-48). For example, in the abstract section (line 46-48),
the author saw dose-dependent changes, which should be mentioned in the conclusion section as well.
Line 383-390: In your schematic presentation, I suggest to align the arrow mark giving final look. For example, the arrow from ROS to IRE1α/TRAF2 can be moved slightly towards left.
An unnecessary bold letter or underline or color can be removed.
Response: We appreciate this comment and Conclusion section in manuscript were revised in blue color. We have revised pictures in the revised version of the Fig 9.

Reviewer 3 Report
In the present manuscript the authors present that H. erinaceus mycelium (HEM) and its ethanolic extract erinacine A have shown protection against oxidative stress induce by MPP+. However the manuscript presents some weakness as detailed below:
They mentioned the isolation of erinacine A from HPLC in a quantitative way. However, they should include the HPLC chromatogram (as supporting material) and the calibration curve used, as well as a LC-MS analysis to be sure that the elucidation of the extract as erinacine A is correct. The treatments with HEM and erinacine A are not clearly expressed for each in vitro cell experiment (before, at the same time or after MPP+), so they need to point out which regimen they have followed in each experiment. The quality of the western blot images need to be improved. From the protein bands they show in the manuscript it is almost impossible to quantify in a proper way the expression levels of the proteins. In figure 1B, the plots shown for the flow cytometry Apoptosis experiments are not the suitable ones for this kind of experiments. The best representation is the FL-1 (Annexin-V) vs. FL-3 (PI) to show the apoptotic and necrotic cells populations under compound's treatments, compared with labelled cells without any treatment. In figures 5, there are some errors in the plot's labels. The methods description need to be improved.Author Response
RE: Revised version of antioxidants-650790
Dear Megan Xu
Assistant Editor,
Enclosed please find one revised version entitled: Post-treatment with erinacine A, a derived diterpenoid of H. erinaceus attenuates neurotoxicity in MPTP model of parkinson's disease, which we would like to submit for publication in Antioxidants.
This revised version has been carefully corrected according editor and referee’s reports point-by-point. We appreciate these valuable comments to strengthen our presentation. Please inform me if any revision is needed. The file marked change in blue color.
Furthermore, I would verify that no part of the manuscript is under consideration for publication elsewhere and it will not submit elsewhere if accepted by Antioxidants and not before the Editorial Office has reached a decision.
Sincerely yours,
Hsing-Chun Kuo, Ph.D.
Associate Professor,
Institute of Nursing and Department of Nursing,
Chang Gung University of Science and Technology,
Chia-Yi Campus, Taiwan.
E-mail: guscsi@gmail.com
TEL: +886-5-3628800
FAX: +886-5-3628866
Response to Reviewer 's Comments
Reviewer(s)' Comments to Author:
Reviewer #3:
Comments and Suggestions for Authors
They mentioned the isolation of erinacine A from HPLC in a quantitative way. However, they should include the HPLC chromatogram (as supporting material) and the calibration curve used, as well as a LC-MS analysis to be sure that the elucidation of the extract as erinacine A is correct. The treatments with HEM and erinacine A are not clearly expressed for each in vitro cell experiment (before, at the same time or after MPP+), so they need to point out which regimen they have followed in each experiment. The quality of the western blot images need to be improved. From the protein bands they show in the manuscript it is almost impossible to quantify in a proper way the expression levels of the proteins. In figure 1B, the plots shown for the flow cytometry Apoptosis experiments are not the suitable ones for this kind of experiments. The best representation is the FL-1 (Annexin-V) vs. FL-3 (PI) to show the apoptotic and necrotic cells populations under compound's treatments, compared with labelled cells without any treatment. In figures 5, there are some errors in the plot's labels. The methods description need to be improved.
Response: We appreciate this remark of reviewer #3’s comment and respond to the comment as follows:
We appreciate this remark of the Editor’s comment.
Chemical compounds suggested in this article Erinacine A (PubChem CID: 10410568), including the HPLC chromatogram (as supporting material) and the calibration curve used, as well as a LC-MS analysis, were shown in Figure 1. These are described in page 15, line 376 to line 380 in the text.We appreciate this comment and we have added to the revised version of the Fig 2 legend. N2a cell line treated with 0.1% DMSO, MPP+ or MPP+ plus Erinacine A (after MPP+ 3hrs, 1 and 5 µM) at 24 h were determined.
In addition to addressing the comment specifically, we have revised Western blots and provide better quality pictures in the revised version of the Figure 3 and Figure 4. Data of Figure 3 and Figure 4 presented in the Western blot are derived from a representative study, and relative quantization are calculated from the duplicate experiments and have considerably improved.
We have revised Flow cytometry analysis and a bar histogram and provide better quality pictures in the revised version of the Fig 2B.
We appreciate this comment and we have added to the revised version of the Fig 5 legend and Fig 5 B, 5C.

Round 2
Reviewer 3 Report
The authors have introduced the modifications recommended during the reviewing process. The new version of the manuscript has more scientific quality and the western blot images now are more easily to interpret and correlate with the statistical analysis performed.
Author Response
Response: We appreciate this remark of reviewer #3’s comment. These are revised in the manuscript and all legends and quantitative analysis of immunoblot with the statistical analysis.
